# Deep Turbulence as a Novel Main Driver for Multi-Specific Toxic Algal Blooms: The Case of an Anoxic and Heavy Metal-Polluted Submarine Canyon That Harbors Toxic Dinoflagellate Resting Cysts

**DOI:** 10.3390/microorganisms12102015

**Published:** 2024-10-04

**Authors:** Camilo Rodríguez-Villegas, Iván Pérez-Santos, Patricio A. Díaz, Ángela M. Baldrich, Matthew R. Lee, Gonzalo S. Saldías, Guido Mancilla-Gutiérrez, Cynthia Urrutia, Claudio R. Navarro, Daniel A. Varela, Lauren Ross, Rosa I. Figueroa

**Affiliations:** 1Centro i~mar, Universidad de Los Lagos, Casilla 557, Puerto Montt 5480000, Chile; ivan.perez@ulagos.cl (I.P.-S.); patricio.diaz@ulagos.cl (P.A.D.); angela.baldrich@ulagos.cl (Á.M.B.); matthew.lee@ulagos.cl (M.R.L.); guido.mancilla@ulagos.cl (G.M.-G.); daniel.varela@ulagos.cl (D.A.V.); 2CeBiB, Universidad de Los Lagos, Casilla 557, Puerto Montt 5480000, Chile; 3Centro de Investigación Oceanográfica COPAS COASTAL, Universidad de Concepción, Concepción 4070386, Chile; gsaldias@ubiobio.cl; 4Centro de Investigación en Ecosistemas de la Patagonia (CIEP), Coyhaique 5950000, Chile; 5Departamento de Física, Facultad de Ciencias, Universidad del Bío-Bío, Concepción 4081112, Chile; 6Departamento de Recursos Naturales y Medio Ambiente, Universidad de Los Lagos, Chinquihue km. 6, Puerto Montt 5480000, Chile; cynthia.urrutia@ulagos.cl (C.U.); claudio.navarro@ulagos.cl (C.R.N.); 7Scientific and Technological Bioresource Nucleus (BIOREN), Universidad de La Frontera, Av. Francisco Salazar 01145, Temuco 4811230, Chile; 8Department of Civil and Environmental Engineering, University of Maine, Orono, ME 04469, USA; lauren.ross1@maine.edu; 9Centro Oceanográfico de Vigo, Instituto Español de Oceanografía (IEO-CSIC), Subida a Radio Faro 50, 36390 Vigo, Spain

**Keywords:** microalgae toxins, submarine canyon, sediment anoxia, dinoflagellate resting cyst, encystment, excystment process

## Abstract

Over the recent decades, an apparent worldwide rise in Harmful Algae Blooms (HABs) has been observed due to the growing exploitation of the coastal environment, the exponential growth of monitoring programs, and growing global maritime transport. HAB species like *Alexandrium catenella*—responsible for paralytic shellfish poisoning (PSP)—*Protoceratium reticulatum*, and *Lingulaulax polyedra* (yessotoxin producers) are a major public concern due to their negative socioeconomic impacts. The significant northward geographical expansion of *A. catenella* into more oceanic-influenced waters from the fjords where it is usually observed needs to be studied. Currently, their northern boundary reaches the 36°S in the Biobio region where sparse vegetative cells were recently observed in the water column. Here, we describe the environment of the Biobio submarine canyon using sediment and water column variables and propose how toxic resting cyst abundance and excystment are coupled with deep-water turbulence (10^−7^ Watt/kg) and intense diapycnal eddy diffusivity (10^−4^ m^2^ s^−1^) processes, which could trigger a mono or multi-specific harmful event. The presence of resting cysts may not constitute an imminent risk, with these resting cysts being subject to resuspension processes, but may represent a potent indicator of the adaptation of HAB species to new environments like the anoxic Biobio canyon.

## 1. Introduction

Oceans and human societies are intimately linked and our health is intrinsically connected to the health of the ocean [1]. Indeed, ocean ecosystems are the most extensive on planet Earth, covering 72% of the surface, so there can be no doubt as to the significance of their effects on the welfare of society, including health, leisure, and the economy [2]. For a comprehensive understanding of the importance of oceans, consider land area within 100 km from the coast and up to 100 m in elevation as a coastal zone; at least ~37.6% of the world’s population live in this zone, which includes a population of 2.15 to 2.90 billion (by distance), plus 0.898 to 1.2 billion (by elevation) living on 9% of the global land area [3].

Coastal zones have experienced the fastest-growing changes due to human population pressures derived from the growth of economic and technological development and sea–land–air interactions, coupled with several oceanic physical processes, all within a climate change framework [2,4]. This environmental and socioeconomic background enhances coastal risks, threatening the ecosystems that provide those economic benefits through habitat conversion, land-cover change, pollutant loads, and the introduction of invasive species [4,5]. In the first case, with one of the largest coasts in the world, the Chilean marine system (covering 161,338 km^2^ of the oceanic continental shelf) faces new and/or unknown biological threats, such as the introduction of new species through invasions or range expansions [5].

This kind of emergent socio-ecosystem threat applies to several species of microalgae responsible for Harmful Algal Blooms (HABs) and High Biomass Blooms (HB-HABs), including phycotoxins and oxygen depletion events. For instance, *Karenia* spp., G. Hansen, and Moestrup (formerly *Gymnodinium*), a dinoflagellate species that forms resting cysts and which has in the past been associated with fish kills in offshore waters, recently formed an intense bloom in the Pitipalena-Añihue Marine Protected Area, located in the inshore waters of the northwestern Patagonia fjords, somewhere where blooms of this species had never previously been recorded [6]. Moreover, human-mediated range extensions of neurotoxic species such as *Alexandrium catenella*, responsible for paralytic shellfish poisoning (PSP), have been associated with the inter-regional movements of shellfish stock and aquaculture equipment [7]. This spreading issue is one of several factors needed for a species to be a successful colonizer, as it can only survive and thrive where the conditions are suitable and remains absent in areas where one or more essential resources or necessary conditions are missing [8]. As a result, each species needs to adapt its growth and reproduction (sexually) in the new environment to be able to persist there, so the environment selects if the spreading individuals can survive within this new environment [9,10].

One way to assess if a dinoflagellate species can be considered a successful colonizer of new areas and potentially become an emergent problem is through the study of recent sediment records in which resting cyst-forming species accumulate. Frequently, as in the case of *A. catenella*, resting cysts are the product of sexual reproduction [11,12,13]. These sediment records also constitute an early warning of potential toxic outbreaks of species that are not detected in the water column and/or allow the identification of toxic species in a new area [14]. This is the case with *Alexandrium catenella*, where its northwards progression has been described based only on the abundance of resting cysts within the Magallanes (55.5° S) and Los Lagos regions (41.5° S) in Chilean Patagonia [15,16]. However, the northward progress of this species has not stopped since the intense bloom of *A. catenella* in 2016. Its presence has been recorded along the Valdivian coast (39.4° S) and, more recently, it was detected in Bahía Coliumo (36° S) in the Biobio administrative region with a scarce density of vegetative cells [17,18]. The main concern is whether the species is present only in a vegetative stage or if the species has established itself in this new geographical area, with sexual compatibility among strains (complex heterothallism), an essential issue in the encystment process [13,19]. If these sexual resting cyst stages of distinct viable cohorts have accumulated in the sediments, it would provide evidence that the species has successfully established in this new geographic area.

Once toxic dinoflagellate resting cysts have been detected in the sediment, it is necessary to determine their viability using an excystment test under controlled conditions. This provides an estimate of what proportion of the cyst population might initiate a harmful bloom in the area. Nonetheless, once the dormancy period has been completed, a mechanism (natural and/or anthropogenic) is needed to disturb the bottom sediments and resuspend the resting cysts accumulated there, exposing them to a change in light, oxygen, and temperature conditions [15,16], which permits the excystment process to occur in the water column or at the sediment surface. Sediment bioturbation [20,21], advective flows, vertical turbulence mixing of water column driven by winds, an upwelling process that favors vertical advection, Ekman transport/pumping [22], and dredging [23,24] have been proposed as natural/anthropogenic mechanisms for resting cyst resuspension. However, the link between these natural/anthropogenic mechanisms with resting cyst resuspension and excystment is very difficult to demonstrate.

Submarine canyons are an ideal ecosystem for exploring resting cyst sinking areas and for testing if toxic species such as *Alexandrium catenella* (among other toxic species) have successfully established in ecological terms (as described above). In this regard, the Biobio submarine canyon (hereafter, BbC), harbors an interesting ecosystem to study. The BbC is located at the northern margin of the distribution of *A. catenella*. It provides between 30 to 60% of nitrates available in the euphotic zone of the coastal waters of the area [25] due to intense onshore and vertical transport [26], support for biologically active zones with high primary productivity, and dense aggregations of fish of commercial interest, favoring the blue economy in the area [27]. Furthermore, the presence of a canyon also provides a sheltered habitat for fish and wildlife during periods of low oceanic productivity [28]. At depth in the canyon, the dissolved oxygen tends to decrease, leading to hypoxic zones [29], which are an ideal environment for the preservation of dinoflagellate resting cysts [20,30,31]. Furthermore, the current asymmetry across the canyon slopes can influence the accumulation zones for organic matter and pollutants such as heavy metals, in addition to the abundance of resting cysts. In this regard, this canyon receives several contaminants of industrial and municipal origin. Additionally, there have been several oil spills since 2000 and effluent discharged from steel and pulp and paper mills entering the Arauco Gulf ([32] and references therein). These kinds of pollutants could negatively affect the metabolic activities of vegetative phytoplankton cells, inhibiting growth and survival rates at determined concentrations [33]. Despite this, some associations were found between the degree of heavy metal pollution and dinoflagellate resting cysts abundance, suggesting that resting cyst production may be a response to heavy metal contamination ([33] and references therein).

In the present study, we assess a new and poorly understood driver for resting cyst resuspension from sediments using a submarine canyon as a case study. Here, we propose deep-water turbulence as a physical mechanism that drives the resting cyst resuspension of three toxic species (*Alexandrium catenella*, saxitoxin (STX) producer, *Protoceratium reticulatum*, and *Lingulaulax polyedra*, yessotoxins (YTXs) producer), which also differed in their responses in controlled excystment tests. Turbulent mixing can become elevated in submarine canyons as a result of upwelling or downwelling conditions due to density currents, oscillatory shelf-break flows, and as a result of internal wave generation or interaction with sharp bathymetry gradients ([34] and references therein). This makes the BbC an excellent case study site to investigate the potential resuspension of resting cysts. The results raise awareness of the likelihood of a multi-specific HAB event in this area.

## 2. Materials and Methods

### 2.1. Survey Area

A submarine canyon consists of a narrow valley that cuts across the continental platform and usually has a “v” shape with steep slopes on both sides [35]. The BbC is one of the most important in Chile due to its dimensions (8.1 km wide and 105 km long) and is located in the Arauco Gulf, on the relatively wide continental shelf off Concepcion [36] (Figure 1A–C). The BbC starts in proximity to the mouth of the Biobio River, orientated from east to west. The canyon meanders with a slope of 1.58° and ends with an abyssal fan with a slope of 2.16° [37]. The presence of this canyon modifies the coastal circulation [38]. Deep water entering the confines of the canyon is transported up onto the continental shelf, contributing to the seasonal upwelling characteristic of the area [36].

The BbC is influenced by two major oceanographic processes reported in the eastern South Pacific Ocean that are related to each other: (1) the Oxygen Minimum Zone (OMZ) and (2) the Humboldt upwelling system. The OMZ has a dissolved oxygen concentration below 20 μM [39], and is the result of high oxygen consumption during organic matter degradation enhanced by the nutrient-rich upwelling water. Additionally, weak water circulation, long residence time, and reduced ventilation enhance oxygen depletion [40,41]. The Humbolt upwelling system is driven by southerly winds along the coast produced by southeast Pacific subtropical anti-cyclones [42,43]. General circulation models predict an intensification of wind-driven coastal upwelling under climate change scenarios [44,45], contributing to OMZ expansion and impacting ecological function [46,47].

During the austral winter, from 27 to 29 July of 2023, recent sediments and water column samples were collected at three sampling stations in a south-to-north meridional transect covering 1.54 km in the BbC. Samples were collected on the southern (36°50′9.60″ S, 73°11′31.20″ W) and northern, (36°49′19.20″ S, 73°11′31.20″ W) slopes and at the base of the canyon (36°49′44.40″ S, 73°11′52.80″ W), respectively (Figure 1A–C). The field sampling procedures are described below.

### 2.2. Water Sampling

#### 2.2.1. Hydrographic and Turbulence Data

At each sampling station, water temperature, salinity, depth, and dissolved oxygen were recorded with a CTDO probe Hydrolab DS5 at a sampling rate of 8 Hz with a descent rate of 1m s^−1^. Turbulence measures were achieved through a vertical micro-profiler model VMP-250 RD (https://rocklandscientific.com/products/profilers/vmp-250, accessed on 27 July 2023). This instrument is equipped with two vertical shear sensors (measuring the mechanical turbulence of the water), which sample at a rate of 512 Hz. Additionally, the VMP-250 was equipped with a high-response RINKO dissolved oxygen sensor. Finally, the turbulence micro-profiler was deployed from the surface to near-bottom, with an optimal vertical free-fall velocity of ~70 cms^−1^. The hydrographic data were then represented graphically using Ocean Data View [48] with a section view of 2.5 km across the Biobiosubmarine canyon and applying Data-Interpolating Variational Analysis (DIVA) gridding software developed by the University of Liège (http://modb.oce.ulg.ac.be/mediawiki/index.php/DIVA, accessed on 29 July 2023).

The dissipation rate of turbulent kinetic energy (*ε*) obtained by the VMP-250 was derived from the vertical shear sensor installed in the micro-profiler, using the following equation: $\varepsilon =7.5\nu {\left(\bar{\frac{\delta u}{\delta z}}\right)}^{2}$, where, *ν* is the kinematic viscosity, *u* is the horizontal velocity, *z* is the vertical coordinate, and therefore ${\left(\bar{\frac{\delta u}{\delta z}}\right)}^{2}$ is the shear variance. The mixing coefficient (*K_shear_*) was calculated in terms of the diapycnal eddy diffusivity [49]. The most frequently used formulation for *K_shear_* estimation is given by $K_{\rho -\mathrm{VMP}}=2\nu$, in which *ν* = 1.9 × 10^−6^ m^2^ s^−1^ and *N* is the buoyancy frequency [50,51,52,53].

#### 2.2.2. Phytoplankton Characterization

Simultaneously, Niskin bottles (5L) were used to collect water samples from six discrete depths (0, 5, 10, 20, 30, and 50 m) for quantitative analyses of phytoplankton above each slope and at the base of the BbC. The samples were immediately fixed with 10 drops of neutral Lugol’s iodine solution (0.5–1% final concentration) [54]. For the quantitative analyses of phytoplankton, 10 mL of the unconcentrated acidic Lugol’s-fixed samples was left to sediment overnight and analyzed with an inverted microscope (Olympus CKX41, Tokyo, Japan) using the method described by Utermöhl (1958) [55]. The detection level using this method was 100 cells L^−1^ (i.e., one cell detected after examination of the entire surface of the sedimentation chamber base plate). Two transects were counted at ×250 magnification to include the smaller and more abundant species. The phytoplankton density was determined to species level when possible using the worldwide-recognized taxonomic guide of Tomas, in addition to a valuable guide from Chilean experts [56,57]. The latter illustrates important morphological features of the Chilean strains within cosmopolitan species or from species of similar latitudes.

### 2.3. Sediment Sampling

At each sampling site, sediment samples were collected using a 0.1 m^2^ Van Veen grab for further identification and quantification of dinoflagellate resting cysts, grain size composition, total organic content (hereafter TOC), and heavy metals (hereafter HM) analysis. Once the grab was onboard, three records of sediment temperature, pH, and redox potential (hereafter redox) were recorded, via the access ports, in haphazard positions using a pre-calibrated portable multi-parameter MultiLine^®^ meter (Multi 3620 IDS probe, WTW, Weilheim, Germany), as detailed in Rodríguez-Villegas et al. [20].

After registering the sediment physical–chemical parameters, three undisturbed sediment sub-samples were obtained from each grab from the first 3 cm using a dark plastic corer of 8 cm in length × 6 cm in diameter, two for dinoflagellate resting cysts analysis and one for HM analysis. We used the upper 3 cm of the sediment samples as this part of the sediment is more susceptible to cyst resuspension events according to Anderson et al. [58], and thus the fraction highly correlated with new bloom occurrence. To avoid triggering resting cyst germination due to oxygen, light, or temperature variations, the air bubbles of each sub-sample were manually removed, wrapped in aluminum foil, and preserved at 4 °C until analyzed, as suggested by Anderson et al. [59]. These procedures induce anoxic conditions due to organism respiration, which are effective in maintaining cyst quiescence [60]. Finally, another 500 cm^−3^ of sediment was collected from the grab to determine TOC and grain size composition.

#### Sediment Grain Size, Total Organic Content, Pigments, and Heavy Metals

Sediment samples were freeze-dried and then analyzed using a series of sieves (2000, 1000, 500, 250, 150, 90, and 63 µm). Each sample was shaken for 15 min in a mechanical sieve shaker (Gilson Company, Inc., Lewis center, OH, USA). The weight of the sediment fraction retained in each sieve and the pan (<63 µm) was then recorded. These data were then used to calculate the granulometry parameters using the “rysgran” R package [61].

Sediment grain size was expressed in values of Phi (Φ), as described in Rodríguez-Villegas et al. [30]. Higher or lower values of Phi (Φ) indicate that sediment samples are composed predominantly of finer or coarser grain sizes, respectively [62]. The proportions of the sediments in the categories of the Φ scale are also presented.

Total organic content (TOC) was estimated by loss on ignition according to the RAMA methodology (Accompanying Resolution N°404 of Environmental Regulation for Aquaculture N°3411/2006). Briefly, sediment samples were air-dried at 60 °C for 72 h. The dry samples were weighed and then placed in a muffle furnace at 450 °C for 6 h to burn off the organic material. The TOC (%) was calculated as the difference between initial and final weight after the 450 °C treatment.

The sediment used for photopigment analysis was first frozen and then freeze-dried (Thermo Savant ModuloD-230, El Nogal, Spain) for approximately 48 h. Analysis of the photopigments in the sediment samples was made using the standard methodology [63]. The sediment samples (2 g) were placed in 50 mL conical tubes with 5 mL of 90% acetone. The tubes were then placed in the fridge in the dark for 12 h at 4 °C. The tubes were then centrifuged at 650 rpm for 10 min. Using a micro-pipette, 280 μL aliquots of each sample were removed from the tubes and placed into the wells of a 96-well microplate; the absorbance of acetone-only blanks in each of the wells was measured prior to the sample analysis. Spectrophotometric measurements were made using a Tecan Infinite200-PRO plate reader to determine photopigment concentrations, with readings taken at the following wavelengths: 631, 647, 663, and 750 nm. The plate was then removed from the reader and 20 μL of 1.2 M HCl was added to each well to acidify the samples. A second set of spectrophotometric measurements was then made at the following wavelengths: 663 and 750 nm. The measured values were then used to calculate Chlorophyll-a and Phaeopigment concentrations.

Heavy metals were processed as described by the Environmental Protection Agency (EPA) according to the 3050b methodology [64]. Each sediment sample was incubated at 60 °C and then sieved with a 60 µm mesh. A total of 1 g from the <60 µm sediment fraction was subjected to acidic digestion using nitric and hydrochloric acid to release the heavy metals from the sediment. Then, 10 mL of hydrogen peroxide [1:1] was used to complete the digestion at 95° ± 5 °C. The remaining solution was analyzed in an atomic spectrophotometer (PERKIN ELMER Inc, ANALYST 200, Waltham, MA, USA) to quantify the heavy metals Cu, Cd, Cr, Pb, Zn, Mn, Ni, Mg, and Fe, all expressed as µg kg^−1^. This analysis was conducted in triplicate.

### 2.4. Sediment Processing, Dinoflagellate Resting Cyst Counts, and Excystment Tests

Resting cysts were extracted from 1 mL of sediment from each sample by sonicating, sieving, and concentrating via centrifugation in a density gradient of colloidal silica (Ludox CLX, Sigma-Aldrich, Darmstadt, Germany), as outlined by Genovesi et al. [65]. Total toxic cysts (*A. catenella*, *P. reticulatum*, and *L. polyedra*) were identified following the taxonomical guide of Matsuoka and Fukuyo (2003) and Anderson et al. (2004) [59,66] and counted in a 1 mL Sedgewick Rafter chamber. Arithmetic mean values with standard deviation are presented for all dinoflagellate cyst abundance.

The controlled excystment tests were carried out using 5 resting cysts from each toxic species. Resting cysts were randomly selected by capillary manipulation using an inverted microscope (Olympus CKX41, Tokyo), as described by Varela et al. (2012) [67]. Each resting cyst was then placed in a single well of a 96-well plate along with 7 drops of filtered seawater (0.22 µm, micropore) with a salinity of 32. The L1 medium was not incorporated based on the observation of Figueroa et al. (2005) [12], where the excystment process for *A. catenella* was observed to be lower under nutrient-replete conditions. Plates were sealed with Parafilm^®^, and water lost due to evaporation was replaced when necessary. Culture plates were then incubated at 12 °C ± 1 °C under an irradiance of 90 µmol photons m^−2^ s^−1^ provided by cool-white fluorescent tubes on a 12:12 h light:dark cycle. The excystment success was monitored daily using microscope observations for each of the three toxic species, for a period of up to 90 days. The median values for excystment for all species are provided.

### 2.5. Data Analysis

A constrained Redundancy Analysis (RDA) was carried out to explore the relationship between sediment physico-chemical predictors such as temperature, pH, redox, TOC, phi, and heavy metals (Cu, Zn Cd, Mg, Ni, Mn, Pb, Fe) with *A. catenella*, *p. reticulatum*, and *L. polyedra* resting cyst abundances using the “tbRDA” function from the “vegan” package from R [68]. This analysis selects a linear combination of the environmental predictors that gives the smallest total residual sum of squares, providing a significance values for each predictor vector using a determination coefficient (R^2^) tested with 10,000 permutations [69]. In addition, a Hellinger transformation was applied to toxic dinoflagellate resting cysts to reduce the importance of highly abundant sampling stations and ensure bias was minimized [70].

Finally, for sediment physical–chemical characteristics, median values with Inter Quartile Range (IQR) are provided, and arithmetic mean values with standard deviation are presented for HM.

## 3. Results

### 3.1. Hydrography and Turbulence

The hydrographic conditions around the BbC denoted cooler (10.5 °C) and denser water (26.6 kg/m^3^) in the deeper area of the canyon (Figure 2A,C). Still, less dense water was registered at the surface layer (0–3 m) due to registering the minimum absolute salinity and maximum water temperature (Figure 2A,B). The warmer (12 °C) and oxygenated (150–200 mM) waters comprised the first 50 m in depth (Figure 2A,D). Moreover, the hypoxic and anoxic layers coincided with the cooler and dense water described below, permitting the identification of the presence of the Equatorial Subsurface water mass (ESSW) inside the canyon (Figure 2D).

The turbulence data obtained with the micro-profiler provide evidence of a high dissipation rate of turbulent kinetic energy within the surface layer ((*ε* = 5.5 × 10^−5^ Watt/kg). Additionally, intense turbulence was registered close to the bottom layer ((*ε* = 7.2 × 10^−7^ Watt/kg) at the BbC southern slope (Figure 2E). Finally, the diapycnal eddy diffusivity calculation produced high values at the surface layer (*K_shear_* = 4.5 × 10^−4^ m^2^/s), where intense turbulence was also registered. In addition, from 50m to the bottom and especially inside the canyon, *K_shear_* was elevated (*K_shear_* = 10^−4^–10^−3^ m^2^/s), indicating the occurrence of intense mixing (Figure 2F).

### 3.2. Phytoplankton Density

The water column samples taken at the three sampling sites contained the following phytoplankton groups: diatoms, dinoflagellates, and silicoflagellates, which were represented by 37, 7, and 1 species, respectively (Table 1). The diatom assemblages were mostly composed of *Chaetoceros* species, and among the toxigenic species, the Domoic Acid (DA) producer *Pseudo-nitzschia australis* was detected from 0–20 m depth at all the sampling sites (Table 1). Within the dinoflagellates, the toxic species *Dinophysis acuminata* (DTX1 and/or PTX2 producer) was notable even when found in low densities and principally towards the southern slope sampling site and in surface waters (0–10 m depth) (Table 1). Moreover, the High Biomass Harmful Algae Bloom (HB-HAB) species *Prorocentrum micans* was observed in surface waters towards the southern slope site as well. However, vegetative cells of HAB resting cyst-forming species such as *Alexandrium catenella*, *Protoceratium reticulatum*, and *Lingulaulax polyedra*, the focus of the present study, were not observed in the water column (Table 1).

### 3.3. Sediment Physical–Chemical Parameters, Phaeopigments, and Heavy Metals

The sediments of the northern slope were coarser than those of the canyon and southern slope, as indicated by the lower values of Φ. The canyon and southern slope both had higher amounts of fine sand compared to the northern slope, which had a large amount of coarse material. The proportions of organic matter in the sediments were also higher in the canyon and southern slope compared to the northern slope (Figure 3A); this same pattern was also observed in the quantities of photopigments in the sediments (Figure 3B). These observations, taken together, suggest that the water current flowing through the canyon is at its strongest along the northern slope, leading to a more depositional environment along the canyon floor and southern slope.

The sediment temperature was similar to that recorded in deep waters, but the sediment was completely anoxic, below −170 mV for all sampling sites, and quite neutral, with pH median values ranging from 7.08 to 7.52 (Figure 4A-C). The heavy metals with the highest concentrations were Fe, followed by Mg and Mn (Figure 5A–C), but the metal concentrations observed in the sediments show little variation between sites (Figure 5). The exception is Mg, which was found at higher concentrations in the sediments of the northern slope and canyon compared to the southern slope (Figure 5A–C).

### 3.4. Toxigenic Dinoflagellate Resting Cysts Abundance

The toxic dinoflagellate resting cysts observed correspond to the saxitoxin and yessotoxin producers *Alexandrium catenella*, *Protoceratium reticulatum*, and *Lingulaulax polyedra*, which were observed to be viable and with their entire cytoplasmatic content, which suggests a good physiological status (Figure 6A–C).

The spatial variability of these toxic dinoflagellate resting cysts showed that the highest resting cyst abundance was recorded in the southern slope of the BbC (Figure 7A–C). Their abundance ranged from 41 to 104 cyst mL^−1^ for *A. catenella*, from 22 to 226 cyst mL^−1^ for *P. reticulatum*, and from 18 to 124 cyst mL^−1^ for *L. polyedra* (Figure 7A–C). This is the first record of *A. catenella* and *L. polyedra* in the Biobio region (36.8° S). No patterns were discernible between toxic resting cyst abundances and heavy metal pollution because the three sampling sites exhibited similar heavy metal concentrations, and the high content of Fe in the sediment samples is likely due to the formation of iron sulfide precipitates under the anoxic conditions.

The excystment test was applied only to samples collected in the southern slope of the BbC as it was the site where resuspension events driven mainly by deep turbulence were most likely to occur. These results showed that resting cyst excystment was 20%, 40%, and 60% for *A. catenella*, *P. reticulatum*, and *L. polyedra*, respectively (Figure 7C). This excystment value was achieved on day 71 (median) for *A. catenella*, day 8 for *P. reticulatum*, and day 4 for *L. polyedra*. This evidence indicates that there exists a high likelihood of a multi-specific HAB event in the area that might threaten the wildlife and marine resources of this ecosystem.

The RDA analysis revealed differences in both the distribution of toxic dinoflagellate resting cysts at each sampling site and the environmental predictors, suggesting that each species was influenced by some environmental drivers (Figure 8). For instance, four environments were detected in the BbC. The first was defined by the correlation between Ni and Cd, mainly associated with the north slope sampling site (top left); the second was delimited by the correlation between sediment temperature and pH, which also is associated with the presence of *L. polyedra* (top right); the third was specified by the correlation of six heavy metals (bottom left) associated with the canyon sampling site, with Ni and Cd influencing the presence of *A. catenella*; and finally, the fourth was defined by the correlation of redox, phi, and TOC (bottom right), which delimited the southern slope sampling site, but included temperature and pH influencing the presence of *P. reticulatum* (Figure 8). All the vectors that define each environmental predictor variable were significant (*p* < 0.05), except redox and Mn (Table 2).

## 4. Discussion

In recent years, an interesting discussion has arisen concerning whether the dinoflagellate resting cysts accumulating on the seafloor are good predictors of future blooms (foremost paradigm stating) or just an indicator of where blooms have already occurred [15,71]. In any case, the resting cysts in sediment records provide valuable information for HAB risk awareness. Indeed, for the Biobio region, this study constitutes the first report of the YTXs producer *L. polyedra* [72] and is also the first time in which *A. catenella* resting cysts were found at their northernmost distribution boundary. The implications of these observations are discussed below, with a focus on *A. catenella*, the main causative species of paralytic shellfish poisoning in Chile and many places around the world [73].

### 4.1. Origin of Alexandrium catenella Resting Cysts: A Successful Colonizer?

The *A. catenella* resting cysts found in the BbC could be a reflection of spreading events associated with shipping activities that connect Chile to a diversity of other parts of the world. The world cargo that is transported transoceanically uses 1.2 × 10^10^ ton^−1^ of ballast water taken on at one location and discharged in another [45,74], with a high likelihood of transporting vegetative/cyst cells to a new area, resulting in biological introduction or spreading events [24,75] along the coast of Chile. Another possible dispersal mechanism could be aquaculture activities, where mussels or equipment containing net cells or cysts are transported from one area to another [7,73]. In addition to those possible mechanisms, vegetative cells of *A. catenella* have been observed in the Biobio region recently [17], and thus the cysts present in the area may represent naturally occurring range expansion under a climate change scenario.

If this is the case, there then exists sexual compatibility and successful sexual reproduction in the *A. catenella* vegetative population, resulting in successful encystment. This is because the Chilean populations are characterized within a complex heterothallic system, given that strains mate only with other compatible strains, with compatibility more complex than a +/− type and compatible pairs differing in their affinity and in their capacity for resting cyst production [76]. In fact, reproductive sexual compatibility has been reported in the Chilean Patagonia, and there are cases in which high reproductive compatibility was found in sympatric locations (e.g., in the Aysén administrative region) [13], as is also possibly the case in the BbC.

The prevailing axiom is that *A. catenella* resting cyst production is usually a response to harsh environmental conditions [10]. However, recent findings (from laboratory tests) suggest that *A. catenella* resting cyst production is modulated both by its physiologically determined sexual affinity and by environmental conditions [77]. For instance, when highly compatible *A. catenella* strains grow with low nutrient concentration, cell density, temperature, and high light intensity, the outcome is a lower abundance of resting cysts (mean: 21 resting cysts mL^−1^). In contrast, when low compatible pairings grow with high nutrient concentration, low cell density and low light intensity, and high temperature, the outcome is a higher abundance of resting cysts (mean: 54 resting cysts mL^−1^) [77]. This evidence suggests that *A. catenella* is an extremely adaptable and resilient species, which makes it a successful colonizer. In this case, the environmental variability in the BbC might not exert a strong negative influence on resting cyst production, and that could explain why, even when scarce vegetative cells have been observed in the area, the conditions are good enough to produce sexual cysts.

On the other hand, the results also suggest that *A. catenella* fulfills the ecological criteria necessary to be considered a successful colonizer of new areas: it is a species with invasiveness and invasibility behavior [78]. This means that *A. catenella* can establish in, spread to, or become abundant in novel communities (invasiveness) and exploit local environmental conditions, allowing for the successful establishment of its propagules (cells and resting cysts) (invasibility). Thus, the complexity of the *A. catenella* life cycle (meroplanktonic strategist) is a crucial factor in allowing the resting cyst formation in both adverse and favorable conditions, enabling it to spread successfully. These results presented here demonstrate that the *A. catenella* population at the northern edge of its range is both dynamic and capable of producing sexual offspring.

### 4.2. Bottom-up and Top-down Regulation of Multi-Specific HABs: The Influence of Deep Turbulence on Resting Cyst Resuspension and Excystment

The resting cysts of *A. catenella* were found in low numbers (max = 104 resting cyst cm^−3^), but this abundance is similar to that recorded in other locations on the continental shelf of northern Chilean Patagonia, where blooms are common and where a maximum of 68 resting cysts cm^−3^ were found recently [22]. However, these abundances contrast with those from other areas of the world, where resting cyst abundances in recent sediments typically range from, for example, 2000 to 6000 resting cysts cm^−3^ [58,79], and in some cases reach >17,000 resting cysts cm^−3^ [80], a fact usually linked to bloom recurrence from resuspended sediments.

In the case of Chilean Patagonia, the concept of high cyst abundances as a necessary precursor to bloom formation may not be appropriate owing to the widespread and dispersed (small and patchy) distribution of the resting cysts [15]. This suggests the likelihood that germination will therefore occur over a large area. However, to date, the germination of those cysts isolated from sediment samples has not been tested, so the results showed that at least 20% of the *A. catenella* resting cysts can develop excystment. This process can be carried out in 71 days, a finding similar to Mardones et al. [19], who tested in the laboratory a minimum dormancy period of 69 days and about 90 days in field conditions [81]. Moreover, the minimum time of mandatory dormancy for *P. reticulatum* and *L. polyedra*, according to laboratory studies, reaches 123 ± 11 days [82] and 60 to 120 days [83], respectively. In this sense, the short excystment time and high excystment rates recorded in this study for *P. reticulatum* and *L. polyedra* (day 8 and 40% and day 4 and 60%, respectively), to the best of our knowledge, are unprecedented. However, precaution should be taken because the cyst dormancy period could be fulfilled prior to the excystment test (in the BbC), so some cysts may be under quiescence and waiting for good conditions to excyst.

These results will not translate into consequences without a bottom-up mechanism that resuspends cysts from sediments, allowing those resting cysts to become a problem once species-specific excystment has occurred. At least along the base and southern slope of the BbC, deep turbulence could transport the resting cysts from the sediments and expose them to conditions above the OMZ, where oxygenated and warmer waters could trigger excystment and fuel the vegetative population in the water column, possibly resulting in a HAB event. Enhanced upwelling events have been observed in the head of the BbC, where near-bottom cold waters can rise more than 100 m [36], in agreement with idealized modeling results [84]. These events in the BbC head could be composed of at least three HAB species from the bottom waters (bottom-up regulated) but could also interact with other HAB and HB-HAB species that were detected in the water column, such as *D. acuminata* (DTX, PTX2 producer), *Pseudonizchia* spp. (DA producer), *and Prorocentrum micans* (High Biomass producer) (top-down regulated). These results suggest that the BbC ecosystem could hold multiple threats from the sediments (according to this study) and from the water column (as previously suggested by Díaz et al. [85]), which highlights the importance of establishing the BbC as a risk zone for HAB outbreaks.

In the Bío-Bío region, HAB events are not as frequent as in the Patagonian fjord system [16]. Nevertheless, the intensity of what has occurred is high in terms of cell density, such as the *Dinophysis acuminata* bloom in spring 2019 in the Gulf of Arauco that reached cell densities of 38.4 × 10^3^ cells L^−1^ in integrated hose samples, generating pectenotoxin-2 (PTX-2) and gymnodimine-A (GYM-A) accumulation in hard razor clams (Tagelus dombeii) [85]. While the presence of *Alexandrium catenella* cysts is the main risk in terms of potential impacts on public health, the presence of *P. reticulatum* cysts is also of great concern. This species has shown an increase in the intensity of events in the last decade in southern Chile [86], with the first sanitary closure due to YTX being recorded in autumn 2022 [87]. In addition, recent work has shown the severe impact of YTXs on larvae and juveniles of filter-feeding bivalves such as the scallops *Argopecten purpuratus* [88,89] and the death of millions of farmed abalone in South Africa due to a severe summer bloom of toxic dinoflagellates *L. polyedra* Pitcher et al., [90]. Thus, the impact of HAB events on artisanal fishing in this area could be high because the benthic resources from the Biobio region are the second most important in the country after Los Lagos in gross catches and in socioeconomic terms, as these activities (e.g., benthic fisheries) in Biobio provide direct employment to more than 14,000 people [91].

The main findings of the present study can be summarized as follows, as detailed in Figure 9.

## 5. Conclusions

The northward expansion of *A. catenella* has successfully established the species in the Biobio region as a population dynamic with sexual offspring capabilities. The toxic dinoflagellates involved in this study, once excystment has taken place, could be resuspended through deep-water turbulence into the water column, which makes blooms in the BbC area a risk. Finally, local hydrographic and environmental factors can greatly affect sexual behavior, which has consequences for the success of encystment and excystment.

## Figures and Tables

**Figure 1 microorganisms-12-02015-f001:**
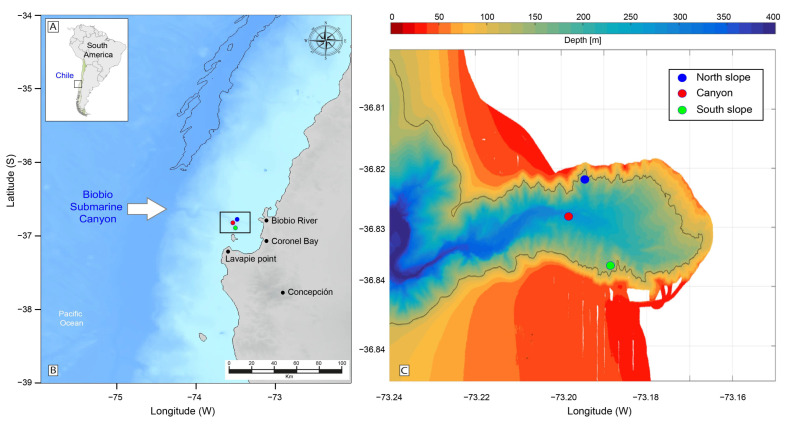
Study area where sediment and water column samples were collected. (**A**) Map of South America showing Chile with the Biobio region (box). (**B**) The specific location of the Biobio submarine canyon. (**C**) A bathymetry map showing the sampling locations within the submarine canyon.

**Figure 2 microorganisms-12-02015-f002:**
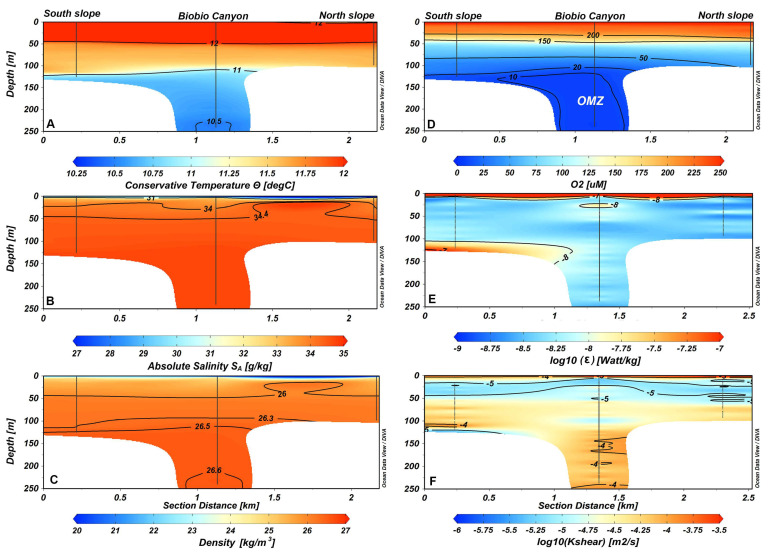
Spatial section of water column variability in the Biobio submarine canyon. (**A**) Conservative temperature [°C]; (**B**) salinity; (**C**) density; (**D**) dissolved oxygen [µM]; (**E**) dissipation rate of turbulent kinetic energy [Watt/Kg]; (**F**) diapycnal eddy diffusivity. The spatial section from left to right displays data on the southern slope, canyon, and northern slope of the Biobio Canyon. (**D**) OMZ, Oxygen Minimum Zone.

**Figure 3 microorganisms-12-02015-f003:**
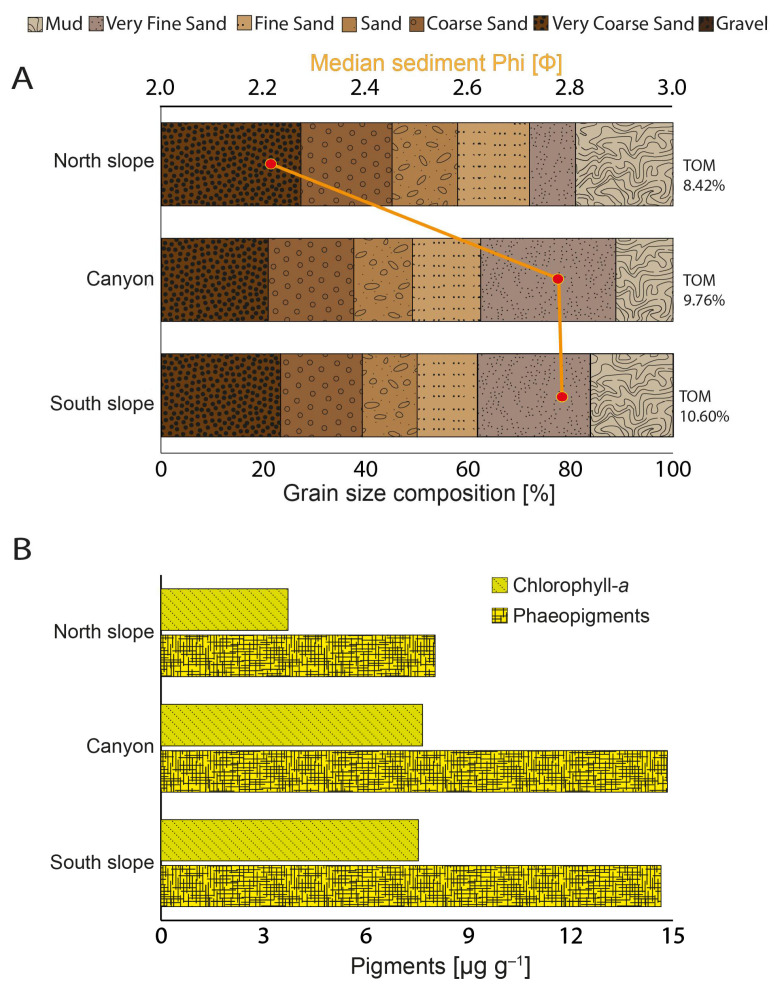
Sediment granulometry and associated pigments in the Biobio submarine canyon. (**A**) Bar plot of the proportions of each sediment size class in each of the sample sites (grain size composition in %), median sediment phi [Φ] (red circles), and Total Organic Matter content (TOM%). (**B**) Concentrations of the photopigments Chlorophyll-*a* and Phaeopigment in the sediment (µg g^−1^) at each of the sample sites.

**Figure 4 microorganisms-12-02015-f004:**
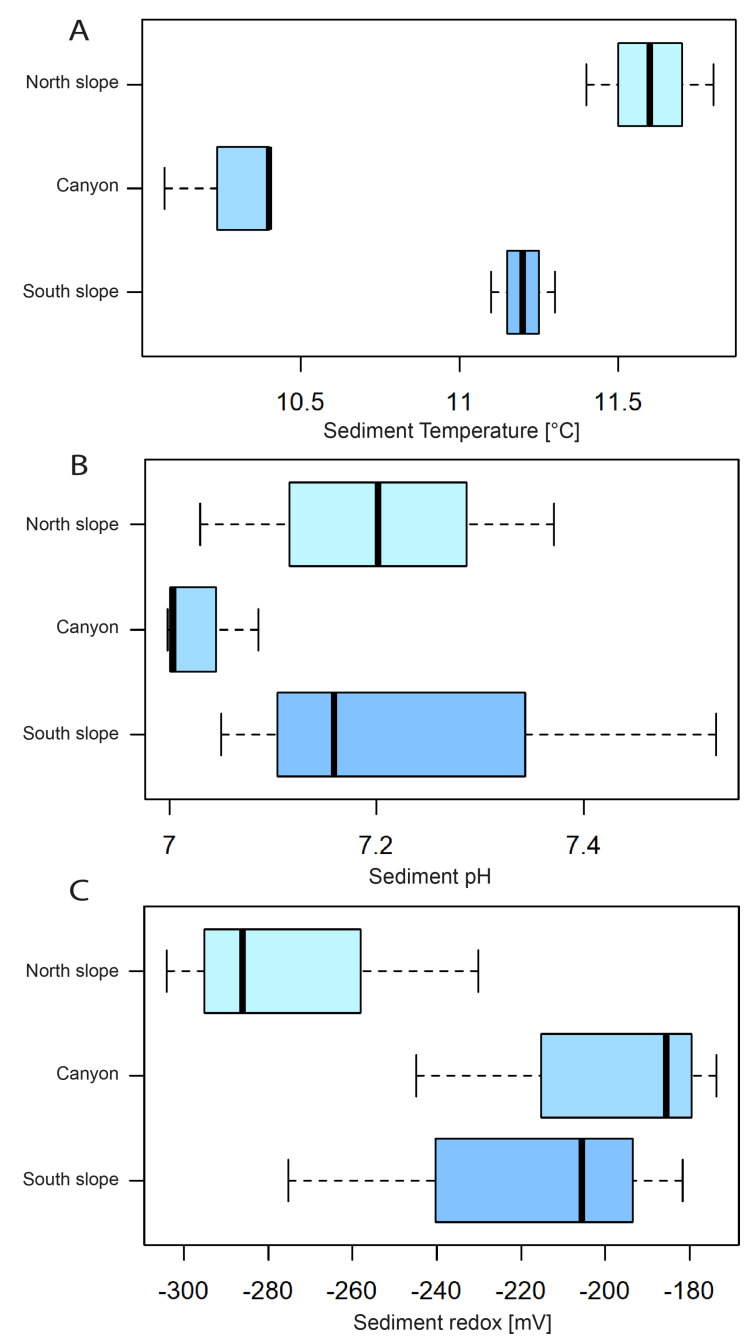
Sediment physical–chemical properties of the northern and southern slopes and the canyon floor of the Biobio submarine canyon. (**A**) Sediment temperature [°T], (**B**) sediment pH, and (**C**) redox potential [mV]. The boxplots show the range (whiskers), median (bold line), and interquartile range (box height).

**Figure 5 microorganisms-12-02015-f005:**
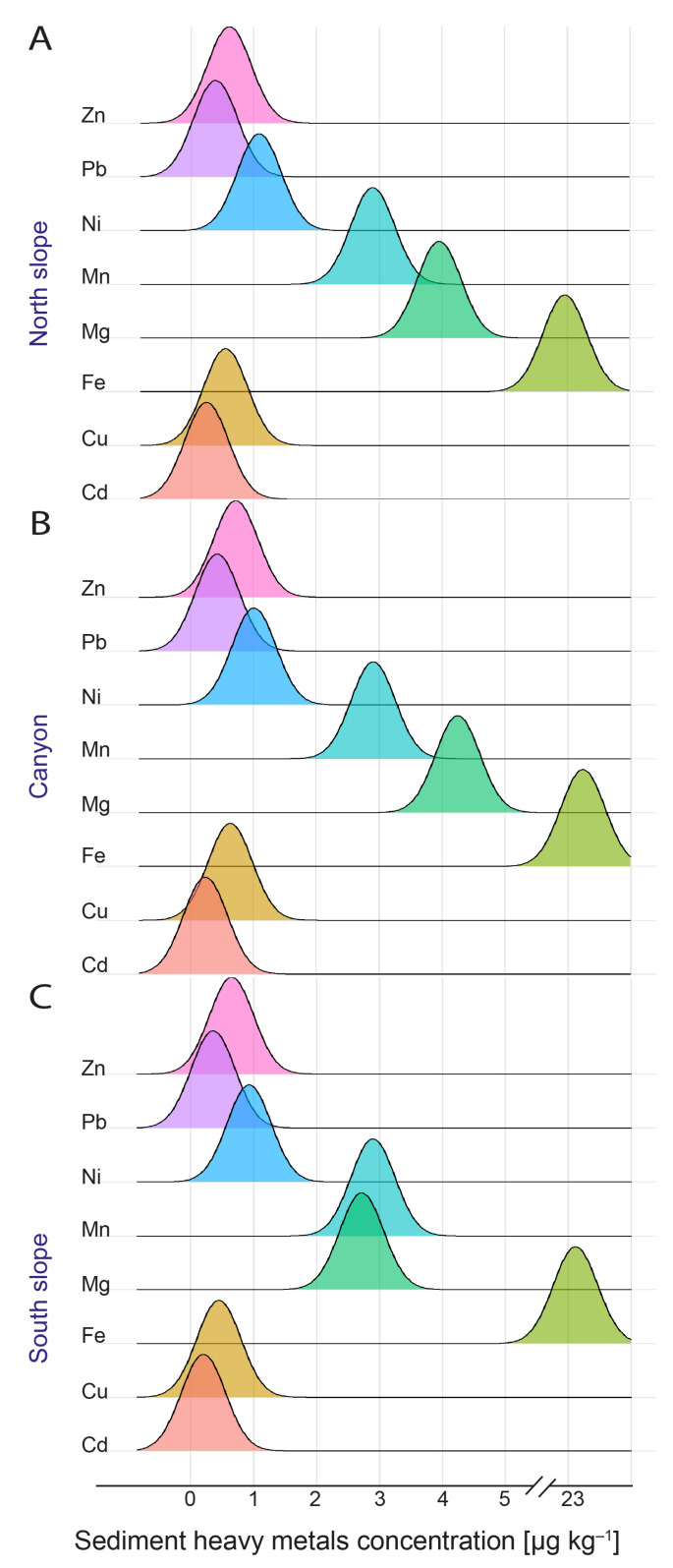
The arithmetic mean of the concentrations of the metals Zinc, Lead, Nickel, Manganese, Magnesium, Iron, Copper, and Cadmium associated with the sediments (µg kg^−1^) at each of the sites of the BbC: (**A**) northern slope, (**B**) canyon, and (**C**) southern slope.

**Figure 6 microorganisms-12-02015-f006:**
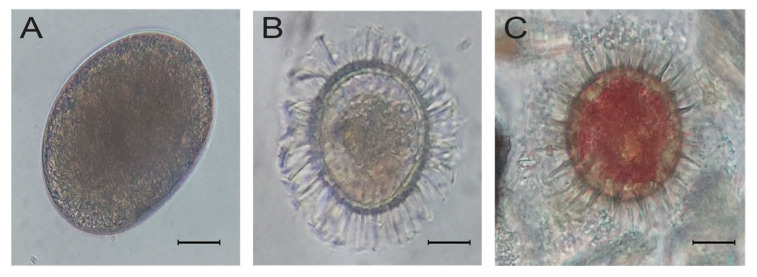
Toxic resting cysts of the dinoflagellates species (**A**) *Alexandrium catenella*, (**B**) *Protoceratium reticulatum*, and (**C**) *Ligulaulax polyedra*, recorded in the BbC. The scale bar is 10 µm.

**Figure 7 microorganisms-12-02015-f007:**
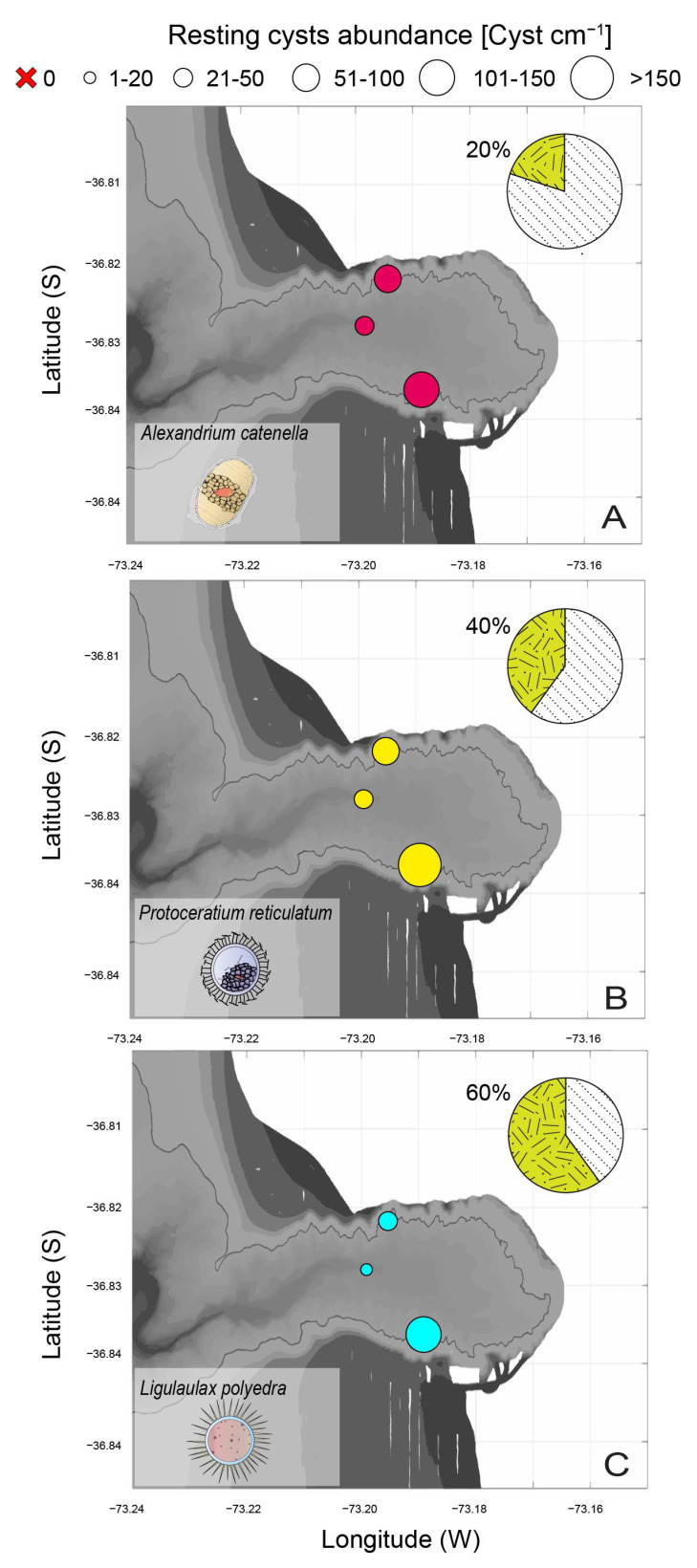
Spatial variability in toxic dinoflagellate resting cysts in the BbC, including cyst germination for each species from the southern slope (given in the pie charts as % germination). (**A**) Cyst abundance of *Alexandrium catenella*, (**B**) cyst abundance of *Protoceratium reticulatum*, and (**C**) cyst abundance of *Lingulaulax polyedra*.

**Figure 8 microorganisms-12-02015-f008:**
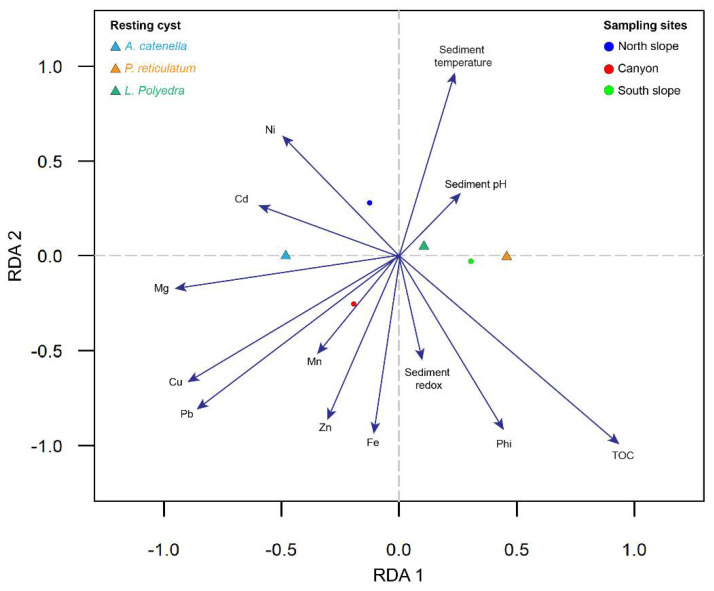
Three plots of Redundancy Analysis (RDA) exploring the relationship between physico-chemical variables and toxic dinoflagellate resting cyst abundance distribution in winter in the BbC.

**Figure 9 microorganisms-12-02015-f009:**
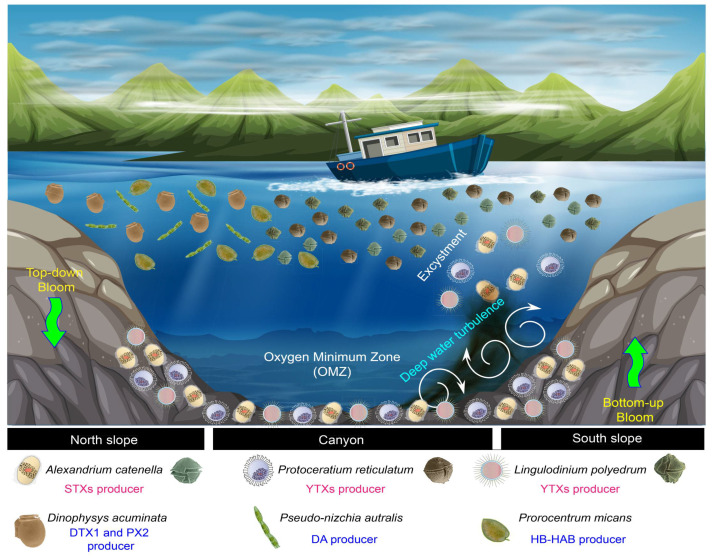
Summary of physical and biological processes involved in top-down and bottom-up bloom dynamics in the BbC.

**Table 1 microorganisms-12-02015-t001:** Phytoplankton assemblages’ density (cells L^−1^) from discrete-depth Niskin bottle sampling in the water column above the north slope, canyon, and south slope of the BbC.

		North Slope						Canyon						South Slope					
	Depth (m)	0	5	10	20	30	50	0	5	10	20	30	50	0	5	10	20	30	50
Group	Species																		
Diatoms	*Actinoptychus senarius*	0	0	600	0	0	0	0	0	0	200	0	0	0	0	0	0	0	0
	*Aulacoseira granulata*	500	700	0	0	0	0	0	0	1300	0	0	0	800	0	0	0	0	0
	*Asterionella formosa*	400	0	0	0	0	0	0	0	0	0	0	0	0	0	0	0	0	0
	*Asterionellopsis glacialis*	600	700	0	400	0	0	200	1000	2100	200	300	0	0	1800	1300	0	0	0
	*Cerataulina pelagica*	0	0	0	0	0	0	0	1000	0	0	0	0	0	0	200	0	0	0
	*Chaetoceros contortus*	0	0	300	0	0	0	0	4400	0	700	0	0	400	1500	0	0	0	0
	*Chaetoceros constrictus*	0	0	0	0	0	0	0	1100	0	0	1400	0	0	600	0	0	0	0
	*Chaetoceros criophilus*	0	0	0	400	0	0	0	300	100	0	0	0	0	0	100	0	0	0
	*Chaetoceros debilis*	0	0	0	0	0	0	0	2500	1800	0	0	0	0	2500	0	0	0	0
	*Chaetoceros diadema*	0	1000	0	0	0	0	0	2200	600	0	0	0	500	300	900	0	0	0
	*Chaetoceros didymus*	0	0	0	0	0	0	0	600	0	0	0	0	0	0	0	0	0	0
	*Chaetoceros lorenzianus*	800	700	0	0	700	0	1400	4200	5600	0	600	0	500	800	0	300	0	0
	*Chaetoceros radicans*	0	0	1400	0	0	0	0	24,400	6200	0	0	0	0	2800	200	0	0	0
	*Chaetoceros socialis*	0	0	0	0	0	0	0	2600	3100	0	0	0	0	300	0	0	0	0
	*Chaetoceros* spp.	0	500	0	400	0	0	0	5300	200	0	300	100	200	3300	100	0	0	200
	*Corethron hystrix*	0	0	0	0	0	0	0	0	200	0	0	100	0	0	0	100	0	0
	*Coscinodiscus* spp.	0	100	100	100	0	0	0	100	200	0	200	0	0	0	0	100	0	0
	*Cylindrotheca closterium*	300	0	100	100	0	0	0	200	0	100	0	0	0	0	0	0	0	0
	*Detonula pumila*	200	2100	0	500	0	0	800	18,700	4000	0	0	0	0	2800	600	0	0	0
	*Eucampia* spp.	0	200	0	0	0	0	0	0	0	0	0	0	0	400	0	0	0	0
	*Fragilaria crotonensis*	0	0	0	0	0	0	0	0	0	0	0	800	400	0	0	0	0	0
	*Guinardia striata*	0	0	0	0	0	0	0	0	500	0	200	0	0	200	0	0	0	0
	*Leptocylindrus danicus*	0	0	0	0	0	0	0	400	0	0	0	0	0	0	0	0	0	0
	*Leptocylindrus minimus*	0	0	0	0	0	0	0	0	200	0	0	0	0	0	0	0	0	0
	*Odontella longicruris*	0	0	0	200	0	100	0	100	100	0	0	0	0	600	200	0	0	0
	*Pleurosigma directum*	0	0	0	0	0	0	0	0	0	0	0	100	0	0	100	200	0	0
	*Pseudo-nitzschia* cf. *australis*	0	3100	0	0	0	0	0	800	0	0	0	0	0	1300	0	200	0	0
	*Pseudo-nitzschia* spp.	100	0	0	200	100	0	0	100	200	400	300	0	600	100	300	100	0	0
	*Rhizosolenia setigera*	0	0	0	0	0	0	0	0	0	0	100	0	0	0	0	0	0	0
	*Rhizosolenia imbricata*	0	100	0	0	0	0	0	0	300	0	0	0	0	0	0	0	0	0
	*Skeletonema* spp.	0	4200	0	0	0	0	7900	5400	2200	300	0	0	3100	8600	3400	0	0	0
	*Stephanopyxis turris*	0	0	0	0	0	0	0	0	0	0	0	0	0	0	100	0	0	0
	*Thalassionema nitzschioides*	1000	27,300	15,000	1900	2900	1100	1300	41,900	31,600	3600	3700	2300	4600	38,400	15,600	2700	800	300
	*Thalassiosira gravida*	0	0	0	0	0	0	100	400	0	0	0	0	200	500	0	0	0	0
	*Thalassiosira subtilis*	0	1100	200	300	0	0	300	19,900	12,800	1600	2800	400	0	5500	3000	3800	800	800
	*Thalassiosira* spp.	300	1900	300	300	200	200	500	7100	2800	300	200	100	200	4300	1100	200	100	200
	*Pennadas*	900	500	700	600	200	200	600	800	2100	1800	1100	200	400	600	500	500	200	100
Dinoflagellates	*Amphidinium* sp.	0	0	0	0	0	0	0	200	0	0	0	0	0	0	0	0	0	0
	*Dinophysis acuminata*	0	0	0	0	0	0	0	100	0	0	0	0	100	200	0	0	0	0
	*Gyrodinium* spp.	100	400	0	0	0	0	0	200	100	0	0	100	200	200	0	0	0	0
	*Prorocentrum micans*	0	0	0	0	0	0	0	0	0	0	0	0	100	0	0	0	0	0
	*Protoperidinium* spp.	0	0	0	100	0	0	0	0	200	0	0	0	100	200	0	100	0	0
	*Torodinium robustum*	0	0	100	0	0	0	0	0	100	0	0	0	0	0	0	0	0	0
	*Tripos pentagonus*	0	100	0	0	0	0	0	0	0	0	0	0	0	0	0	0	0	0
Silicoflagellates	*Dictyocha speculum*	0	0	0	0	0	0	0	0	100	0	0	0	0	0	0	0	0	0
Others	*Mesodinium rubrum*	0	0	0	0	0	0	0	0	300	100	0	0	100	100	0	0	0	0

**Table 2 microorganisms-12-02015-t002:** Predictor vector variables for toxic dinoflagellates resting cyst abundance in the BbC used in the RDA analysis.

Vectors	RDA 1	RDA 2	R^2^	Pr (>R)
Temperature	0.320	0.947	0.940	0.0041
pH	0.734	0.6783	0.293	0.3601
Redox	0.154	−0.988	0.440	0.1912
TOC	0.723	−0.690	1.000	0.0044
Phi	0.321	−0.946	1.000	0.0043
Cu	−0.935	−0.354	0.999	0.0042
Zn	−0.265	−0.964	0.994	0.0047
Cd	−0.972	0.234	0.966	0.0061
Mg	−0.997	−0.075	1.000	0.0042
Ni	−0.827	0.562	0.996	0.0048
Mn	−0.562	−0.826	0.504	0.152
Pb	−0.895	−0.444	0.995	0.0046
Fe	−0.036	−0.999	0.998	0.0040

## Data Availability

The original contributions presented in the study are included in the article. Further inquiries can be directed to the corresponding authors.

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
