# Peer review of "Deep Turbulence as a Novel Main Driver for Multi-Specific Toxic Algal Blooms: The Case of an Anoxic and Heavy Metal-Polluted Submarine Canyon That Harbors Toxic Dinoflagellate Resting Cysts"

_microorganisms, 2024, doi:10.3390/microorganisms12102015_

Round 1
Reviewer 1 Report
Comments and Suggestions for Authors
The manuscript entitled “Deep turbulence as a novel main driver for multi-specific toxic algal blooms: the case of an anoxic and heavy metal polluted submarine canyon that harbors toxic dinoflagellate resting cysts» is devoted to Harmful Algae Blooms (HAB) which is an important problem due to its negative socioeconomic impacts. Authors describe the environment (sediment and water column) of the Biobio submarine canyon in the eastern South Pacific Ocean near Chile and propose how toxic dinoflagellate resting cyst (Alexandrium catenella, Protoceratium reticulatum, and Lingulodinium polyedra) abundance can case imminent risk due to deep water turbulence. To my mind this manuscript is topical and corresponding to the aims and scopes of the “Microorganisms” journal. I am ready to recommend it for publication after correcting several comments.
1. It is worth indicating the GPS coordinates of the sampling and indicating the distance between the sampling points.
2. Methods for identifying organisms and assessing their exposure to metals are not well described.
3. The authors should indicate how representative the three samples they collected are for understanding the problem and formulating global conclusions.
4. For the journal microorganism, I would advise the authors to add the key table with the diversity of phytoplankton from the supplementary to the main text
5. As advice on the diversity of microorganisms, I would like to supplement the light microscopy data with 18SrRNA analysis data,
6. In my opinion, one table would be enough to describe the parameters pH, Eh and metal content.
7. The story with heavy metals in this manuscript seems completely unnecessary to me. The authors should describe their origin in the introduction and formulate more clearly the purpose of this work, taking into account the role of heavy metals. They simply exist and for some reason the toxicity analysis was carried out, by the way, using not entirely clear methods.
8. I do not quite understand how the data shown in Figure 7 were obtained.
9. The reasoning about the source of Alexandrium catenella resting cysts in paragraph 4.1 is not quite clear. Should they not be there? Are they not specific to these habitats?
10. In my opinion, one of the important factors of blooming - temperature - should be used in the discussion. It is worth using data on the temperature regime for a time period and discussing it as a factor in the development of the studied organisms. Are there any real data on phytoplankton blooming at the studied sampling points?
11. The conclusion for such difficult-to-understand material should provide answers to questions about the feasibility of all the experiments conducted. It should be significantly expanded and supplemented with possible forecasts, etc.
Reviewer 2 Report
Comments and Suggestions for Authors
The data presented in this manuscript is of scientific interest due to the significance of toxic algal blooms in marine ecosystems. The manuscript is prepared well with a clear introduction, detailed methodology, explanation of results and a concise discussion. The figures are clear and easy to follow. I didn't find any technical issues in methodology section. However, I would like to advise the authors to consider following points in the final draft.
1. Please include an appropriate statistical analysis for sediment characteristics, heavy metal and plankton data to see any significance difference between the sampling stations.
Round 2
Reviewer 1 Report
Comments and Suggestions for Authors
The authors have significantly improved the manuscript, taking into account all my comments. I believe the manuscript can be published in this form.